# Clinical and genetic spectra in patients with dystrophinopathy in Korea: A single-center study

UnKyu Yun[1], Seung-Ah Lee[2], Won Ah Choi[3], Seong-Woong Kang[3], Go Hun Seo[4], Jung Hwan Lee[5], Goeun Park[6], Sujee Lee[6], Young-Chul Choi[2], Hyung Jun Park[2]*

1 Department of Neurology, Bucheon Sejong Hospital, Bucheon, Korea, 2 Department of Neurology, Rehabilitation Institute of Neuromuscular Disease, Gangnam Severance Hospital, Yonsei University College of Medicine, Seoul, Republic of Korea, 3 Department of Rehabilitation, Rehabilitation Institute of Neuromuscular Disease, Gangnam Severance Hospital, Yonsei University College of Medicine, Seoul, Republic of Korea, 4 Division of Medical Genetics, 3billion, Inc. Seoul, Republic of Korea, 5 Department of Neurology, Seoul St. Mary's Hospital, College of Medicine, The Catholic University of Korea, Seoul, Republic of Korea, 6 Biostatistics Collaboration Unit, Yonsei University College of Medicine, Seoul, Republic of Korea

* hjpark316@yuhs.ac

**Data Availability Statement:** All relevant data are within the manuscript and its Supporting Information files.

**Funding:** This study was supported by a new faculty research seed money grant of Yonsei

## Abstract

Dystrophinopathy is a group of inherited phenotypes arising from pathogenic variants in *DMD*. We evaluated the clinical and genetic characteristics of Korean patients with genetically confirmed dystrophinopathy. We retrospectively reviewed medical records (January 2004-September 2020) from the myopathy database maintained at the study hospital and found 227 patients from 218 unrelated families with dystrophinopathy. Clinical phenotypes included 120 (53%) Duchenne muscular dystrophy (DMD) cases, 20 (9%) intermediate phenotype muscular dystrophy (IMD) cases, 65 (29%) Becker muscular dystrophy (BMD) cases, 18 (8%) undetermined phenotypes, and 4 (2%) symptomatic carriers. The median ages at symptom onset and diagnosis were 5.0 years (interquartile range [IQR]: 3.8–8.0) and 12.0 years (IQR: 7.0–21.0), respectively. Total manual muscle test (MMT) scores decreased annually in patients with DMD, IMD, and BMD. Overall, when age increased by 1 year, total MMT scores decreased on average by -1.978, -1.681, and -1.303 in patients with DMD (p<0.001), IMD (p<0.001), and BMD (p<0.001), respectively. Exonic deletion and duplication were reported in 147 (67%) and 31 (14%) of the 218 unrelated probands, respectively. A total of 37 different small sequence variants were found in 40 (18%) of the 218 probands. The reading frame rule was applicable to 142 (94%) of the 151 probands. The present results highlight the long-term natural history and genetic spectrum of dystrophinopathy in a large-scale Korean cohort.

## Introduction

Dystrophinopathy is a group of inherited phenotypes arising from pathogenic variants in *DMD* [1, 2]. *DMD* is one of the largest human genes, comprising 79 exons and 7 promoters,

University College of Medicine for 2020 (3-2020-0127). The funder provided support in the form of salaries for authors [S.A.L., W.A.C., S.W.K., G.P, S.L. Y.C.C. and H.J.P], but did not have any additional role in the study design, data collection and analysis, decision to publish, or preparation of the manuscript. The specific roles of these authors are articulated in the 'author contributions' section.

**Competing interests:** Go Hun Seo is employed by the company 3billion. This does not alter our adherence to PLOS ONE policies on sharing data and materials. The remaining authors declare that the research was conducted in the absence of any commercial or financial relationships that could be construed as a potential conflict of interest.

with more than 2.5 million base pairs on chromosome Xp21.2 [3–5]. More than 7,000 different variants in *DMD* have been reported to date (https://databases.lovd.nl/shared/genes/DMD). The most common changes in *DMD* are exonic deletions, accounting for 69% of all dystrophin mutations. Exonic duplications occur in approximately 11% of patients; the remaining 20% are small sequence variants [6].

Dystrophinopathy is the most common type of muscular dystrophy in children, with a prevalence of 21.2/100,000 in school-aged boys [7]. It is characterized by progressive muscle degeneration and weakness. The clinical spectrum of dystrophinopathy is highly variable, presenting as severe Duchenne muscular dystrophy (DMD), milder Becker muscular dystrophy (BMD), X-linked dilated cardiomyopathy, and symptomatic female carriers: these clinical phenotypes appear to be related to the amount of dystrophin in skeletal muscle [8, 9]. Patients with dystrophinopathy typically experience muscle weakness, along with cardiac, respiratory, and orthopedic complications of varying degrees. Currently, there are no curative therapies for dystrophinopathy. However, multidisciplinary care planning, including corticosteroids, cardiac medications, orthopedic surgery, rehabilitation, and assisted ventilation, has been found to improve quality of life and clinical outcomes, delaying death in patients with DMD into their 30s or 40s [10, 11].

Recently, various therapies based on molecular mechanisms, including gene therapy, cell therapy, read-through drugs, and antisense oligonucleotides for exon-skipping, have been studied and subjected to clinical trials: of these, four antisense oligonucleotide drugs (eteplirsen, golodirsen, viltolarsen, and casimersen) have been approved by the United States Food and Drug Administration for use in treating dystrophinopathy [12–15]. As such, genetic diagnosis of dystrophinopathy is becoming increasingly important. However, there are few studies on the pathological and genetic characteristics of dystrophinopathy in Korea [16–18].

To evaluate genotype-phenotype correlations in dystrophinopathy, we investigated the genetic spectrum and changes in clinical and laboratory findings with disease progression in Korean patients with genetically confirmed dystrophinopathy.

## Materials and methods

### Study participants

The present study hospital is one of the largest myology centers in Korea, and our database contains information on 2,331 unrelated patients with myopathy. We reviewed medical records from January 2004 to September 2020 from the myopathy database and identified 227 patients from 218 unrelated families with dystrophinopathy. This research protocol was approved by the institutional review board of Gangnam Severance Hospital, Korea (IRB No: 3-2020-0127). The need for written informed consent was waived by the board because this was a retrospective study.

### Phenotype and laboratory assessment

We analyzed the clinical spectrum of patients with dystrophinopathy using their medical records, which included information on age at symptom onset, sex, family history, motor weakness, presence of scoliosis, loss of ambulation, respiratory discomfort, dilated cardiomyopathy, and use of mechanical ventilation. Five subgroups were defined, including DMD, BMD, intermediate phenotype muscular dystrophy (IMD), undetermined phenotype (UD) (patients who were ambulatory on medical records, but had no records after the age of 12 years) and symptomatic carriers (females with muscle weakness of any severity). DMD, BMD, and intermediate phenotype muscular dystrophy (IMD) were defined according to age at loss of ambulation (DMD <13 years, BMD ≥16 years, and 13 years ≤ IMD <16 years) in accordance with

established diagnostic criteria [19]. Motor weakness in individual movements was assessed using manual muscle testing (MMT) scores (0, 1, 2, 3-, 3, 3+, 4-, 4, 4+, 5-, and 5). Next, MMT scores were converted to an 11-point scale from 0 to 10. A total MMT score was calculated as the sum of 10 strength values, which included the strength values of shoulder abduction, elbow extension, elbow flexion, wrist extension, wrist flexion, hip flexion, knee extension, knee flexion, ankle dorsiflexion, and ankle plantarflexion. The results of prior creatine kinase (CK), echocardiography studies, and muscle biopsies were extracted from the medical records in 227, 162, and 43 patients, respectively.

## Genetic spectrum

Among the 218 unrelated probands, 178 with exonic deletions or duplications were diagnosed with multiplex ligation-dependent probe amplification (MLPA). The remaining 40 with small sequence variants were genetically confirmed by targeted sequencing or whole-exome sequencing. All identified variants were classified into benign, likely benign, uncertain significance, pathogenic, or likely pathogenic variants, according to the guidelines of the American College of Medical Genetics and Genomics [20]. The numbering for the pathogenic variants of *DMD* was based on the cDNA sequence (accession: NM_004006.2). Next, we checked the reading frame rule in 151 probands with exonic deletion or duplication using the LOVD exonic deletions/duplications reading–frame checker (https://www.dmd.nl/). The reading frame rule was not evaluated in 27 unrelated probands with exonic deletion or duplication for the following reasons: (1) 16 probands had UD due to incomplete clinical data, (2) probands had exonic duplication affecting two regions, (3) 3 probands were symptomatic carriers, (4) 2 probands had exonic deletions including a translation initiation site, and (5) 2 probands had exonic deletions and duplications including the translation termination site.

## Statistical analysis

Descriptive statistics of the clinical spectrum of patients are presented as frequencies and percentages for categorical variables and as means and standard deviations for continuous variables. To compare the degree of muscle weakness and serum levels of CK between groups over time, time effects, group effects, and their interactions were included in a linear mixed model. A first-order autoregressive covariance structure was considered such that the larger the interval between time points of repeatedly measured data, the less of a correlation there was. All statistical analyses were performed using SAS version 9.4 (SAS Institute, Cary, NC, USA). The significance level for the statistical analyses was set at $P < 0.05$.

## Results

### Clinical spectrum

Table 1 and S1 Table summarize the clinical spectrum data of 227 Korean patients from 218 unrelated families with dystrophinopathy. There were 223 males (98%) and 4 females (2%). Clinical phenotypes included 120 (53%) DMD, 20 (9%) IMD, 65 (29%) BMD, 18 (8%) UD, and 4 (2%) symptomatic carrier cases. A family history was positive in 70 (31%) patients. The median ages at symptom onset, diagnosis, and last examination were 5.0 (interquartile range [IQR]: 3.8–8.0), 12.0 (IQR: 7.0–21.0), and 23.0 (IQR: 17.0–31.0) years, respectively. The median follow-up period was 9 (IQR:3.0–15.0) years. Muscle biopsies were performed in 43 (19%) of the 227 patients. Among them, analysis of nine samples indicated end-stage muscle disease. The remaining 34 muscle samples showed many degenerative/regenerative muscle fibers, infiltration of inflammatory cells, and increased endomysial fibrosis.

**Table 1. Clinical characteristics of 227 Korean patients with dystrophinopathy.**

| | Total patients | DMD | IMD | BMD | UD | Symptomatic carrier |
|---|---|---|---|---|---|---|
| Number of patients | 227 | 120 (53%) | 20 (9%) | 65 (29%) | 18 (8%) | 4 (2%) |
| Age at the symptom onset, year | 5.0 [3.8–8.0] | 5.0 [3.0–6.0] | 5.5 [4.0–7.8] | 10.0 [6.0–15.5] | 4.0 [3.0–6.3] | 17.0 [5.8–29.0] |
| Age at the diagnosis, year | 12.0 [7.0–21.0] | 9.0 [7.0–15.8] | 12.0 [8.0–22.3] | 20.0 [13.0–28.5] | 5.5 [3.8–7.0] | 33.5 [12.3–36.0] |
| Age at the last follow-up, year | 23.0 [17.0–31.0] | 22.5 [17.0–28.0] | 30.5 [24.3–35.5] | 27.0 [19.0–37.5] | 7.0 [5.8–8.3] | 33.5 [12.3–42.8] |
| Family history | 71 (31%) | 37 (31%) | 6 (30%) | 25 (38%) | | |
| Males | 223 (98%) | 120 (100%) | 20 (100%) | 65 (100%) | 18 (100%) | 0 (0%) |
| MMT-sum score at the last follow-up | 14.0 [22.0–61.0] | 15.0 [12.3–22.0] | 15.0 [12.5–19.5] | 70.0 [39.5–90.0] | 81.5 [64.0–100.0] | 83.0 [37.8–90.0] |
| Scoliosis | | | | | | |
| Scoliosis | 146 (64%) | 109 (91%) | 19 (95%) | 15 (23%) | 3 (17%) | 0 (0%) |
| Scoliosis surgery | 33 (15%) | 31 (26%) | 2 (10%) | 0 (0%) | 0 (0%) | 0 (0%) |
| Age at scoliosis surgery | 14.0 [13.0–17.0] | 14.0 [13.0–17.0] | 12.0 and 18.0 | – | – | – |
| Loss of ambulation | 154 (68%) | 120 (100%) | 20 (100%) | 13 (20%) | 0 (0%) | 0 (0%) |
| Age at the loss of ambulation | 11.0 [10.0–12.0] | 10.0 [9.0–11.0] | 13.5 [130–14.0] | 19.0 [17.0–25.0] | – | – |
| Respiratory function | | | | | | |
| Respiratory discomfort | 121 (53%) | 93 (78%) | 17 (85%) | 11 (17%) | 0 (0%) | 0 (0%) |
| Number of patients with ventilator use | 111 (49%) | 87 (72%) | 16 (80%) | 8 (12%) | 0 (0%) | 0 (0%) |
| Age at the start of mechanical ventilation | 21.0 [18.0–25.0] | 20.0 [17.0–22.0] | 26.0 [23.0–28.0] | 37.5 [30.3–44.8] | – | – |
| Cardiac evaluation | n = 162 | n = 105 | n = 20 | n = 36 | n = 1 | n = 1 |
| Number of patients with DCMP | 77 (48%) | 57 (54%) | 8 (40%) | 12 (33%) | 0 (0%) | 0(0%) |
| Age at the diagnosis of DCMP | 23.0 [18.0–28.0] | 21.0 [17.5–26.0] | 27.5 [20.8–28.0] | 34.0 [26.5–38.0] | – | – |
| Death until the last follow-up | 7 (3%) | 6 (5%) | 1 (5%) | 0 (0%) | 0 (0%) | 0 (0%) |
| Serum CK level at the last follow-up | 1,034.0 [375.3–3,310.3] | 646.5 [275.8–2,033.0] | 411.0 [259.0–866.0] | 1,865.0 [777.0–3,288.0] | 14,898.0 [10,998–17,231] | 2,165.0 [1,195.0–2165.0] |

DMD, Duchenne muscular dystrophy; BMD, Becker muscular dystrophy; IMD, intermediate phenotype muscular dystrophy; UD, undetermined phenotype; DCMP, dilated cardiomyopathy; CK, creatine kinase; MMT, manual muscle test: the Medical Research Council 5-point scale for strength was converted to an 11-point scale (0, 1, 2, 3-, 3, 3+, 4-, 4, 4+, 5-, and 5). The observed MMT scores ranged from 0 to 10 for each movement assessed. The MMT-sum score was the sum of 10 strength values, including those for shoulder abduction, elbow extension, elbow flexion, wrist extension, wrist flexion, hip flexion, knee extension, knee flexion, ankle dorsiflexion, and ankle plantarflexion.

Immunohistochemical analysis of the N-terminal, rod-domain, and C-terminal of dystrophin protein showed a total loss of expression in 20 patients with DMD, IMD, or UD and patchy and faint staining of sarcolemma in 14 patients with BMD.

Among all patients, 18 were classified as having UD phenotype. The results of muscle biopsy were extracted from medical records in only one patient (ID16). Therein, immunohistochemistry for the rod domain, C-terminus, and N-terminus of dystrophin protein showed a total loss of expression. All four symptomatic carriers complained of asymmetric motor weakness of the proximal leg muscles; however, they did not complain of cardiac, respiratory, or orthopedic complications. Therefore, we only compared the clinical characteristics of patients with DMD, IMD, and BMD. Total MMT scores decreased annually in patients with DMD, IMD, and BMD (Fig 1). Overall, when age increased by 1 year, total MMT scores decreased on average by -1.978, -1.681, and -1.303 in patients with DMD ($p<0.001$), IMD ($p<0.001$), and BMD ($p<0.001$), respectively. The slope difference between DMD and BMD was 0.675; this difference was significant ($p<0.001$). However, the slope was not significantly different

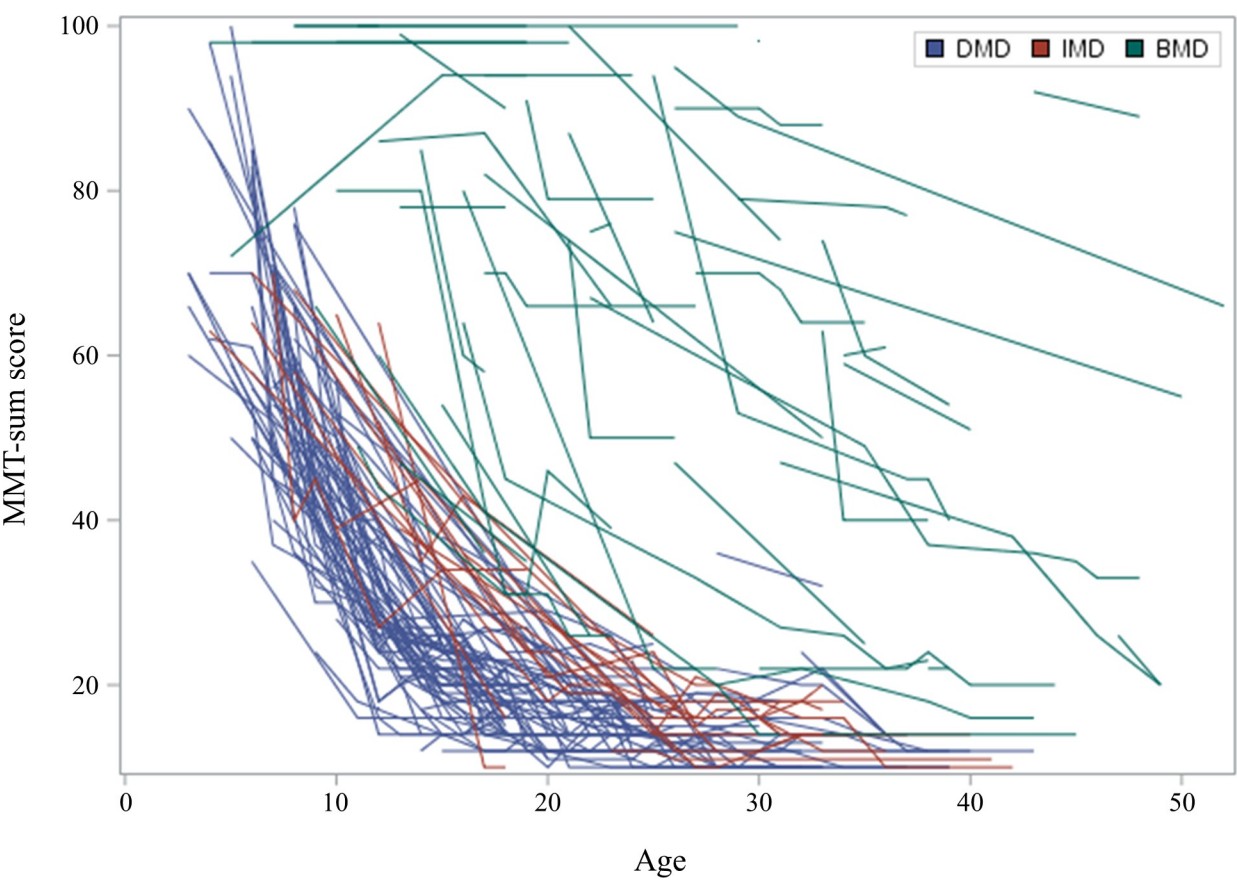

**Fig 1. Changes in manual muscle test sum scores with age: A spaghetti plot.** Individual changes in patients with Duchenne muscular dystrophy (DMD), intermediate phenotype muscular dystrophy (IMD), and Becker muscular dystrophy (BMD). Blue, red, and green colors indicate patients with DMD, IMD, and BMD, respectively. *MMT, manual muscle test: the MMT for strength was converted to an 11-point scale (0, 1, 2, 3-, 3, 3+, 4-, 4, 4+, 5-, and 5). The observed MMT scores ranged from 0 to 10 for each movement assessed. The total MMT score represents the sum of 10 strength values, including those for shoulder abduction, elbow extension, elbow flexion, wrist extension, wrist flexion, hip flexion, knee extension, knee flexion, ankle dorsiflexion, and ankle plantarflexion.

between DMD and IMD (p = 0.217) or between IMD and BMD (p = 0.150). Median serum CK levels at the time of diagnosis and the last follow-up were 3,262 (IQR: 1,132–12,052) and 1,034 (IQR: 375–3,310), respectively. Average serum CK levels decreased at an annual rate in patients with DMD, IMD, and BMD (Fig 2). When age increased by 1 year, serum CK levels decreased on average by -633.497, -356.871, and -127.657 in patients with DMD (p<0.001), IMD (p<0.001), and BMD (p = 0.039), respectively. This annual decrease in serum CK levels was significantly steeper in patients with DMD than in patients with IMD (p = 0.003) and BMD (p<0.001). The median ages at the last examination were 22.5 [IQR: 17.0–28.0], 30.5 [IQR: 24.3–35.5], and 27.0 [IQR:19.0–37.5] years in 120 patients with DMD, 20 patients with IMD, and 65 patients with BMD, respectively. Scoliosis was significantly more frequent in patients with DMD (109 patients, 91%) and IMD (19 patients, 95%) than in patients with BMD (15 patients, 23%, p<0.001). Loss of ambulation was significantly more frequent in patients with DMD (118 patients, 98%) and IMD (21 patients, 100%) than in patients with BMD (14 patients, 21%, p<0.001). Mechanical ventilation use was also significantly more frequent in patients with DMD (86 patients, 71%) and IMD (17 patients, 81%) than in patients with BMD (8 patients, 12%, p<0.001).

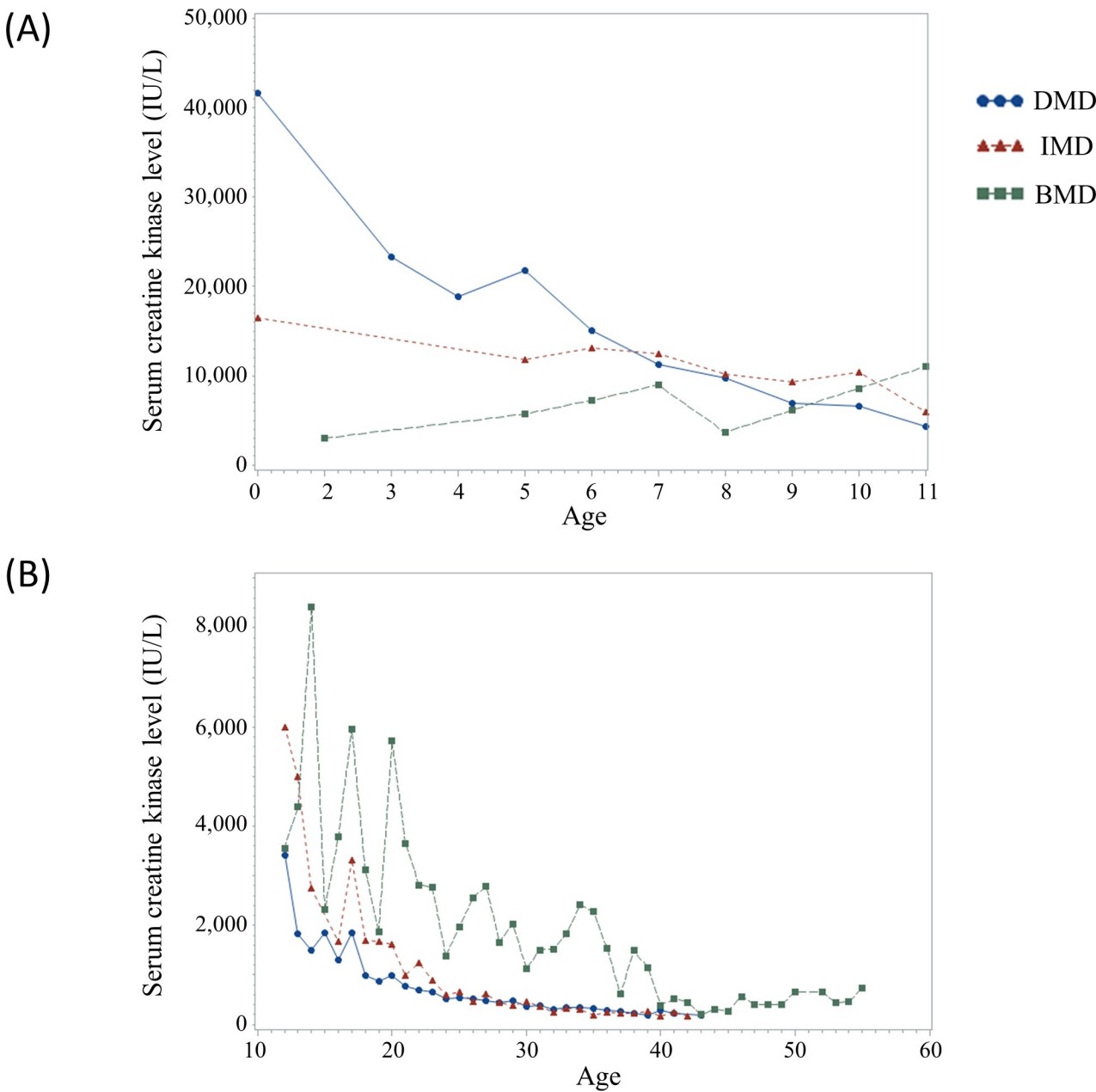

**Fig 2. Changes in mean serum creatine kinase (CK) levels with age.** Blue, red, and green colors indicate patients with Duchenne muscular dystrophy (DMD), intermediate phenotype muscular dystrophy (IMD), and Becker muscular dystrophy, respectively. The average annual decrease in serum creatine kinase level was significantly steeper in patients with DMD than in patients with IMD (p = 0.003) and BMD (p<0.001). (A) Change in mean serum CK level with age for ages 0–12 and (B) change in mean serum CK level with age for ages 12 and above.

### Genetic spectrum

Exonic deletions and duplications were recorded in 147 (67%) and 31 (14%) of the 218 unrelated probands, respectively. The most common genotype was a deletion of exons 45–47 in 17 (8%) probands, followed by deletion of exons 45–48 in 7 (3%) probands and deletion of exon 51 in 7 (3%) probands (Table 2). The frequencies of exonic deletions (93, 69%) in 134

**Table 2. Relationship between common exonic deletion/duplication and clinical phenotypes in Korean patients with dystrophinopathy.**

| Exonic deletion/duplication | Total | DMD | IMD | BMD | UD | Symptomatic carrier | Reading-frame |
|---|---|---|---|---|---|---|---|
| Deletion of exons 45–47 | 17 | 0 | 0 | 14 | 3 | 0 | In-frame |
| Deletion of exons 45–48 | 7 | 0 | 0 | 7 | 0 | 0 | In-frame |
| Deletion of exon 51 | 7 | 6 | 1 | 0 | 0 | 0 | Out-of-frame |
| Deletion of exons 3–7 | 6 | 2 | 0 | 3 | 1 | 0 | Out-of-frame |
| Deletion of exon 45 | 6 | 5 | 1 | 0 | 0 | 0 | Out-of-frame |
| Deletion of exons 49–50 | 6 | 3 | 1 | 0 | 2 | 0 | Out-of-frame |
| Deletion of exons 45–49 | 6 | 0 | 0 | 6 | 0 | 0 | In-frame |
| Deletion of exons 45–50 | 5 | 4 | 0 | 0 | 1 | 0 | Out-of-frame |
| Deletion of exons 46–47 | 5 | 5 | 0 | 0 | 0 | 0 | Out-of-frame |
| Deletion of exons 46–48 | 5 | 4 | 0 | 0 | 0 | 1 | Out-of-frame |
| Deletion of exon 44 | 4 | 3 | 0 | 0 | 1 | 0 | Out-of-frame |
| Deletion of exons 45–52 | 4 | 3 | 1 | 0 | 0 | 0 | Out-of-frame |
| Deletion of exons 45–55 | 4 | 0 | 0 | 13 | 1 | 0 | In-frame |
| Deletion of exons 48–52 | 4 | 3 | 0 | 0 | 0 | 1 | Out-of-frame |
| Deletion of exons 45–54 | 3 | 1 | 0 | 1 | 1 | 0 | Out-of-frame |
| Deletion of exons 46–52 | 3 | 2 | 1 | 0 | 0 | 0 | Out-of-frame |
| Deletion of exon 50 | 3 | 3 | 0 | 0 | 0 | 0 | Out-of-frame |
| Deletion of exons 50–52 | 3 | 3 | 0 | 0 | 0 | 0 | Out-of-frame |
| Duplication of exon 2 | 3 | 3 | 2 | 0 | 1 | 0 | Out-of-frame |

DMD, Duchenne muscular dystrophy, BMD, Becker muscular dystrophy, IMD, intermediate phenotype muscular dystrophy, UD, undetermined phenotype.

probands with DMD or IMD were similar to those of exonic deletion (41, 65%) in 63 probands with BMD. However, the frequency of small sequence variants (19, 14%) in 134 probands with DMD or IMD was lower than that (14, 22%) in 63 probands with BMD. Exonic deletions starting in the proximal hot spot (exons 2–20) and the distal hot spot (exons 45–55) accounted for 22 (14%) and 104 (72%) of the 147 regions, respectively (Fig 3A). Exonic deletions affecting both regions were not found. Exonic duplications affecting the two regions were found in four probands with DMD. Four patients with DMD (ID 139, ID 153, ID187, and ID 201) had duplications of exons 2–7 and 45–51, duplications of exons 5–37 and 50–59, duplications of exons 52–53 and 56–61, and duplications of exons 52–53 and 56–61, respectively (S1 Table). Therefore, we found 35 regions of exonic duplications in 31 probands. Exonic duplication starting in the distal hot spot (exons 45–55) and the proximal hot spot (exons 2–20) accounted for 22 (63%) and 7 (20%) of the five regions, respectively (Fig 3B). A total of 37 different small sequence variants were found in 40 (18%) of the 218 unrelated probands. There were 16 different nonsense variants in 18 (8.3%) unrelated probands, 10 splicing-site variants in 11 (5.0%), nine frameshift variants in 9 (4.1%), one deep-intronic variant in 1 (0.5%), and one missense variant in 1 (0.5%) (Fig 4). Among the sequence variants, four variants were novel (namely, c.491delCinsTT, c.3374C>A, c.4171A>T, and c.8219_8228delACCTCCAAGG).

## Exonic deletions rescuable through mono-exon skipping

The most common exon skips applicable to the largest number of patients were skipping of exon 51 in 18 (13%) of 134 probands with exonic deletion, followed by skipping of exon 45 in 16 (12%), exon 53 in 13 (10%), exon 44 in 13 (10%), exon 43 in 7 (5%), exon 50 in 7 (5%), and exon 53 in 7 (5%) probands.

(A)

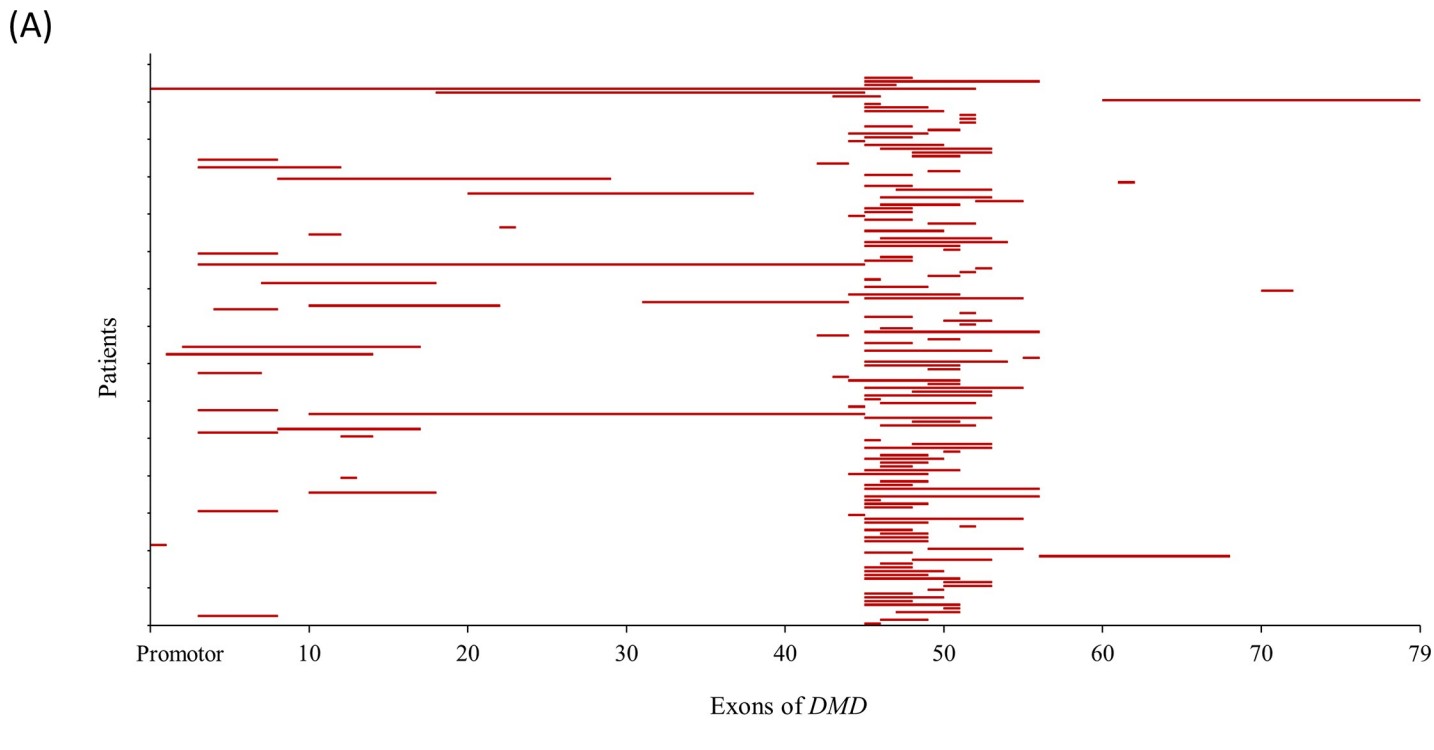

(B)

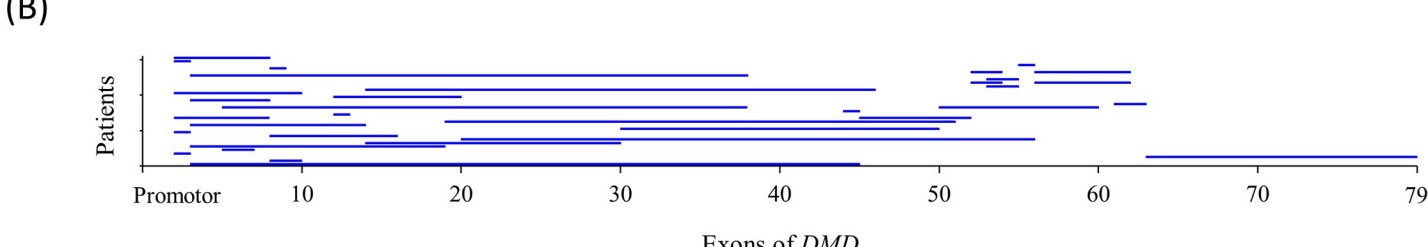

**Fig 3. Distribution of large rearrangements in 178 probands with dystrophinopathy.** Red (A) and blue (B) colors indicate exonic deletions and exonic duplications, respectively.

### Genotype-phenotype correlations

We evaluated the reading frame rule in 151 probands with exonic deletions or duplications. Among the 151 probands, 106 (70%) had out-of-frame variants, whereas 45 (30%) had in-frame variants. The reading frame rule was applied to 142 (94%) probands. Among them, 102 (96%) of 106 probands with out-of-frame variants had a severe phenotype (DMD or IMD), and 40 (89%) of the 45 probands with in-frame variants had a mild phenotype (BMD). Nine unrelated probands did not follow the reading frame rule. Among the four probands with BMD and an out-of-frame deletion, three patients had a deletion of exons 3–7 and one had a deletion of exons 45–54. Four patients with DMD had in-frame deletion of exons 10–44, 49–51, and 56–67 and in-frame duplication of exons 30–49. One patient with IMD had in-frame duplication of exons 3–18. Further, out-of-frame deletion of exons 3–7 was found in two patients with DMD and three patients with BMD (Table 2). Among the sequence variants, nonsense variants were frequently found in 13 (10%) of 134 probands with DMD or IMD, compared with 4 (7%) of 63 probands with BMD. However, splicing variants were frequently found in 6 (10%) of 63 probands with BMD, compared with 6 (4%) of 134 probands with DMD or IMD.

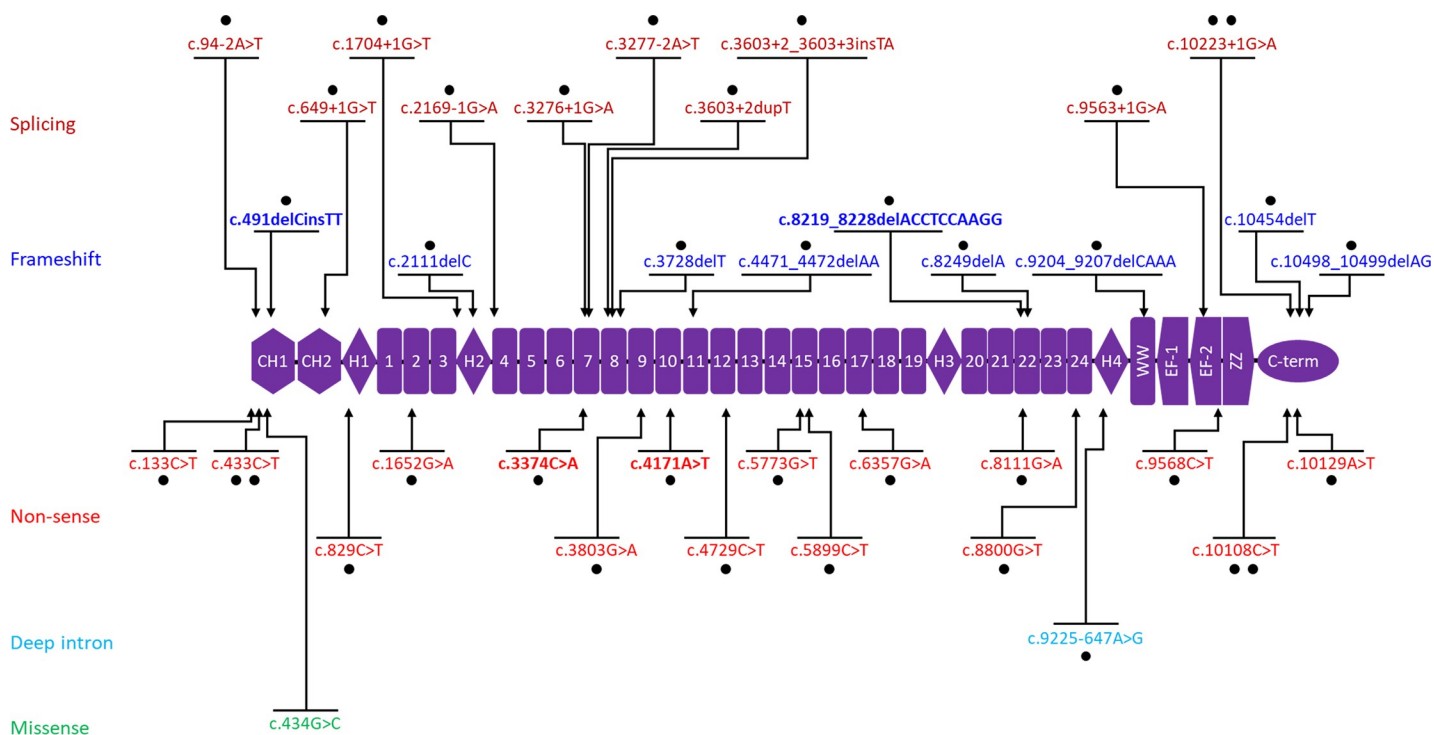

**Fig 4. Small sequence variants in *DMD* in 40 probands with dystrophinopathy.** Red, blue, pink, light blue, and green colors indicate splicing, frameshift, non-sense, deep intron, and missense variants, respectively. Bold text indicates novel variants. Dots indicate the frequency of each variant.

## Discussion

The present study outlines the clinical and genetic spectra of 227 Korean patients from 218 unrelated families with dystrophinopathy. All patients were genetically diagnosed with dystrophinopathy through MLPA or next-generation sequencing.

The clinical presentation of our patients included progressive skeletal muscle weakness, elevated serum CK levels, scoliosis, dilated cardiomyopathy, and respiratory discomfort. Most patients with DMD were non-ambulatory at the age of 10, used a mechanical ventilator at the age of 20, and were diagnosed with dilated cardiomyopathy at the age of 21 years. Total MMT scores and serum CK levels decreased more rapidly with age in patients with DMD than those with IMD or BMD. These findings were compatible with those in previous reports [3, 21, 22].

Our study showed that the proportions of large deletions, large duplications, and small sequence variants were 67%, 14%, and 18%, respectively. This is consistent with previous research [6, 18, 23, 24]. Our study showed that there were exonic deletions and exonic duplications mainly in the distal and proximal hot spots, respectively. The small sequence variants were evenly distributed throughout the entire gene. These results are also compatible with previous results [18, 19, 23, 25]. Further, we found that skipping of exons 51, 45, and 53 could be beneficial to Korean patients with dystrophinopathy. These results indicate that currently developed exon skipping therapies (eteplirsen, golodirsen, viltolarsen, and casimersen) can help many Korean patients with DMD [12, 13].

Regarding the relationship between genotype and phenotype, the reading frame rule showed 94% agreement with the DMD global database [6]. The most inconsistent variant of the reading frame rule was the deletion of exons 3–7, which is an out-of-frame variant; however, it was found in only two patients with DMD and three patients with BMD. The

relationship between this variant and BMD is well known [26]. However, the mechanisms of other inconsistent large rearrangements of the reading frame rule are not clear.

This study has several limitations. First, it was a retrospective study based on medical records. Therefore, we could not include various clinical scales, such as a 6-min walk and the North Star Ambulatory Assessment. We could only analyze limited clinical scales, such as MMT scores. Second, our study was not a multicenter cohort study. Third, this study only included Korean patients. Fourth, we could not analyze the effects of steroids on patients with dystrophinopathy in this study. The use of steroids prolongs the ambulatory phase of dystrophinopathy. Therefore, the lack of steroid information may result in errors in distinguishing DMD from IMD and BMD. However, we could not confirm the exact drug history, including steroids, because that most of patients had also been treated at other hospitals. We could only estimate that about two-thirds of patients had been treated with steroids according to the experience of other myology centers in Korea [18].

The present results highlight the long-term natural history and the genetic spectrum of a large-scale Korean cohort of patients with dystrophinopathy. We demonstrated that the phenotype-genotype relationship and skipping of exons 51, 45, and 53 could be utilized for the management of Korean patients with dystrophinopathy. Our results provide insights for developing therapeutic drugs and designing clinical trials in Korean patients with dystrophinopathy.

## Supporting information

**S1 Table. Clinical features of 227 patients from 218 unrelated families with dystrophinopathy.**
(PDF)

## Acknowledgments

The authors would like to thank the patients and their families for their help with this work.

## Author Contributions

**Conceptualization:** UnKyu Yun, Young-Chul Choi, Hyung Jun Park.

**Data curation:** UnKyu Yun, Seung-Ah Lee, Won Ah Choi, Seong-Woong Kang, Go Hun Seo, Jung Hwan Lee, Goeun Park, Sujee Lee, Hyung Jun Park.

**Formal analysis:** Won Ah Choi, Jung Hwan Lee, Young-Chul Choi, Hyung Jun Park.

**Funding acquisition:** Hyung Jun Park.

**Methodology:** Go Hun Seo, Jung Hwan Lee.

**Supervision:** Young-Chul Choi.

**Visualization:** Goeun Park.

**Writing – original draft:** UnKyu Yun, Hyung Jun Park.

**Writing – review & editing:** Seung-Ah Lee, Young-Chul Choi, Hyung Jun Park.

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
