## [Decision Letter · Decision Letter 0]

11 Jun 2021

PONE-D-21-16348

Clinical and genetic spectra in patients with dystrophinopathy in Korea: A single-center study

PLOS ONE

Dear Dr. Park,

Thank you for submitting your manuscript to PLOS ONE. After careful consideration, we feel that it has merit but does not fully meet PLOS ONE’s publication criteria as it currently stands. Therefore, we invite you to submit a revised version of the manuscript that addresses the points raised during the review process.

Both reviewers ask for a number of additional pieces of information as well as some editorial changes.  Please address these, either by making changes, or by providing an explanation.

We look forward to receiving your revised manuscript.

Kind regards,

Michael Kyba

Academic Editor

PLOS ONE

Journal Requirements:

"NO authors have competing interests"

We note that one or more of the authors are employed by a commercial company: 3bilion, Inc..

2.1. Please provide an amended Funding Statement declaring this commercial affiliation, as well as a statement regarding the Role of Funders in your study. If the funding organization did not play a role in the study design, data collection and analysis, decision to publish, or preparation of the manuscript and only provided financial support in the form of authors' salaries and/or research materials, please review your statements relating to the author contributions, and ensure you have specifically and accurately indicated the role(s) that these authors had in your study. You can update author roles in the Author Contributions section of the online submission form.

2.2. Please also provide an updated Competing Interests Statement declaring this commercial affiliation along with any other relevant declarations relating to employment, consultancy, patents, products in development, or marketed products, etc.  

Reviewers' comments:

Reviewer's Responses to Questions

**Comments to the Author**

1. Is the manuscript technically sound, and do the data support the conclusions?

Reviewer #1: Yes

Reviewer #2: Yes

2. Has the statistical analysis been performed appropriately and rigorously? 

Reviewer #1: Yes

Reviewer #2: Yes

3. Have the authors made all data underlying the findings in their manuscript fully available?

Reviewer #1: Yes

Reviewer #2: Yes

4. Is the manuscript presented in an intelligible fashion and written in standard English?

Reviewer #1: Yes

Reviewer #2: Yes

5. Review Comments to the Author

Reviewer #1: This is a solid, well-written manuscript describing the clinical and genetic characteristics of a large single center cohort of dystrophinopathy patients from Korea. This information will be valuable if the authorities in Korea are considering regulatory and access issues for the new molecular therapies for this disease. I have the following recommendations to improve the manuscript.

1. I recommend alternate references to replace #1 and #2. The authors are correct in citing Louis Kunkel, as his laboratory was responsible for the initial cloning of DMD and discovering the protein dystrophin, but the seminal articles are: (1) Monaco AP et al, Nature 1986;323:646-650 (initial cloning of DMD); (2) Koenig M et al, Cell 1987;50:509-517 (cloning of entire DMD gene); and Hoffman EP et al, Cell 1987;51:919-928 (initial discovery of the protein dystrophin). Some or all of these articles should be cited instead.

2. Regarding the life expectancy into the 30s and 40s for DMD, that is correct but only in the context of good supportive care. Without such care, the life expectancy is much shorter and thus this caveat should be noted.

3. The Introduction notes 2 antisense oligonucleotide drugs but the Discussion mentions 4. There are 4 such compounds approved in the US to date so the Introduction should be corrected.

4. I am a little confused about the composition of the UD cohort. The Methods section suggests that they are presymptomatic patients, and in the Results it suggests that the UD cohort has incomplete medical records. One or both of these passages should be clarified for consistency.

5. It is best to avoid the term "wheelchair-bound", it is better to say "non-ambulatory" instead.

6. In the methods, it is best not to say "We also assessed the serum creatine kinase..." and "Echocardiography was performed..." as the wording suggests that these data were collected prospectively. As this is a retrospective study, it is better to say that results of prior CK and echocardiography studies were extracted from the medical records.

7. In the Genetic spectrum section, the first sentence should end with ", respectively."

8. The description of the 4 exonic duplications affecting two regions. I assume these are large duplications rather than two duplications for each patient but this should be clarified, perhaps even by indicating the start and end exons for the duplications.

9. The genotype-phenotype correlation data is in line with the findings of other studies so I believe this information is largely correct. However, given that steroids prolong the ambulatory phase of the disease, the lack of steroid information potentially introduces errors as a patient who would ordinarily be classified as DMD may walk a bit past the age of 16 with steroid therapy. If possible, it would help to obtain steroid information. If this is not possible, at the very least this limitation should be expanded in the Discussion and also information should be provided on how widespread steroid use is in the DMD population in Korea.

10. In Figure 2, it is difficult to interpret the exact CK levels at the later age ranges. It might be more informative to show two separate graphs, one for ages 0-12 or so and the other for ages 12 and above. Thus the second graph would be capped at 10,000 and it will be more clear what the later numbers are.

11. Figure 3 would be more informative if bars or lines were shown indicating the actual rearranged DNA segments.

Reviewer #2: The Authors report on 227 Korean patients from 218 unrelated families with dystrophinopathy; they describe retrospectively the clinical course of the disease. In all the patients they performed manual muscle testing (MMT) scores and CK dosage, only some patients underwent Echocardiography and muscle biopsy.

In all the patients was identified a pathogenic mutation in the dystrophin gene.

Clinical findings are compatible with those of previous reports; the Authors include an additional phenotype named undetermined phenotype (UD) for the patients who were not wheelchair-bound and too young to be defined as DMD or BMD. Regarding this group, in the supplementary table 1 18 patients are reported as “clinical phenotype pending”. Is the UD phenotype?

I think the group UD is confounding, if the muscle biopsy has been performed I would suggest to define the phenotype as BMD or DMD based on the immunohystochemical analysis ; if not performed I would classify these patients referring to the phenotype expected based on the genotype. All the 18 mutations are reported in the LOVD pages and only the phenotype associated to the duplication of exon 14-45 it could be difficult to predict ( transcript analysis is necessary)

A part form this group, it would be interesting to add some information on the muscle biopsy performed in 43 patients and to correlate it with the genotype.

Regarding the genetic data the frequencies of exonic deletions/duplications and small mutations are similar to the ones previously reported in other countries. Do the Authors have cases of DMD confirmed at the biopsy ( absence of dystrophin) and negative to molecular analysis?

This paper has some limitations as described by the Authors but it is important because is the first report on dystrophinopathies in Korean patients and give data for developing therapeutic drugs and designing clinical trials.

I have few comments :

Lane 37 “dystrophinopathies is a genetic muscle disease” Is not correct, is a group of inherited phenotypes all due to dystrophin mutations.

ABSTRACT and Lane 58 instead of “sequence variants” I’d rather say “mutations”

Lane 62 1% of deep intronic mutations is missing

Lane 65 is the most common type of muscular dystrophy I would add “IN CHILDREN”

Lane 105 “undetermined phenotype (UD) (for the patients who were not wheelchair-bound and too young to be defined as DMD or BMD) “ it is not specificed the age considered as cut off

Lane 219 “Nine unrelated probands did not follow the reading frame rule” A part from the patients with the del 3-7 for which the explanation is known, in the other patients it would be worth to study the transcript if the RNA from muscle biopsy is available.

6. PLOS authors have the option to publish the peer review history of their article (what does this mean?). If published, this will include your full peer review and any attached files.

Reviewer #1: **Yes: **Peter B. Kang

Reviewer #2: No

---

## [Author Response · Author response to Decision Letter 0]

28 Jun 2021

Dear Editor,

Thank you for your letter regarding our manuscript “Clinical and genetic spectra in patients with dystrophinopathy in Korea: A single-center study” and the Reviewers’ comments. 

We have made the necessary corrections in accordance with the suggestions of the reviewers, and provided replies to the reviewers on how we have responded to their points in the manuscript. The changes made have been highlighted in yellow.

We believe that the comments have improved the quality of our manuscript, and we hope you will find our revised manuscript acceptable for publication.

Sincerely yours,

Hyung Jun Park

Hyung Jun Park, M.D., Ph.D., Department of Neurology, Gangnam Severance Hospital, Yonsei University College of Medicine, Seoul, Korea, 211 Eonju-ro, Gangnam-gu, Seoul 06273, Korea. Tel: +82-2-2019-3329, Fax: +82-2-3462-5904, E-mail: hjpark316@yuhs.ac

 

[Editor]

Comment 1: Please ensure that your manuscript meets PLOS ONE's style requirements, including those for file naming. The PLOS ONE style templates can be found at

 Response 1: Thank you for your comment. We have changed the corresponding sentences and words according to the PLOS ONE style templates.

Comment 2: Thank you for stating the following in the Competing Interests section:

"NO authors have competing interests" We note that one or more of the authors are employed by a commercial company: 3bilion, Inc..

Response 2: Thank you for important comment. The COI section has been changed as follows:

Lines 290-294: Competing interests 

Go Hun Seo is employed by the company 3billion. This does not alter our adherence to PLOS ONE policies on sharing data and materials. The remaining authors declare that the research was conducted in the absence of any commercial or financial relationships that could be construed as a potential conflict of interest.

Comment 3: Please provide an amended Funding Statement declaring this commercial affiliation, as well as a statement regarding the Role of Funders in your study. If the funding organization did not play a role in the study design, data collection and analysis, decision to publish, or preparation of the manuscript and only provided financial support in the form of authors' salaries and/or research materials, please review your statements relating to the author contributions, and ensure you have specifically and accurately indicated the role(s) that these authors had in your study. You can update author roles in the Author Contributions section of the online submission form. Please also include the following statement within your amended Funding Statement.

Response 3: Thank you for your important comment. Even though Go Hun Seo is employed at the company, the company did not play a role in the study design, data collection and analysis, decision to publish, or preparation of the manuscript. We have also added the author contributions as follows.

Lines 284-288: The funder provided support in the form of salaries for authors [S.A.L., W.A.C., S.W.K., G.P, S.L. Y.C.C. and H.J.P], but did not have any additional role in the study design, data collection and analysis, decision to publish, or preparation of the manuscript. The specific roles of these authors are articulated in the ‘author contributions’ section. 

Lines 300-303: Author contributions

Concept and study design: Y.C.C. and H.J.P. Data acquisition and analysis: U.K.Y., S.A.L., W.A.C., S.W.K., G.H.S., J.H.L., G.P., S.L., and H.J.P. Drafting the manuscript and figures: U.K.Y., S.A.L., G.P., S.L., and H.J.P. Revision: S.A.L. and H.J.P.

Comment 4: Please include both an updated Funding Statement and Competing Interests Statement in your cover letter. We will change the online submission form on your behalf.

Response 4: Thank you for your comment. We have added an updated Funding Statement and Competing Interests Statement in the cover letter. 

Comment 5: Please include captions for your Supporting Information files at the end of your manuscript, and update any in-text citations to match accordingly. Please see our Supporting Information guidelines for more information: http://journals.plos.org/plosone/s/supporting-information.

Response 5: Thank you for your comment. We have changed the corresponding sentences and words according to the Supporting Information guidelines. 

[Reviewer 1]

Reviewer #1: This is a solid, well-written manuscript describing the clinical and genetic characteristics of a large single center cohort of dystrophinopathy patients from Korea. This information will be valuable if the authorities in Korea are considering regulatory and access issues for the new molecular therapies for this disease. I have the following recommendations to improve the manuscript.

Comment 1: I recommend alternate references to replace #1 and #2. The authors are correct in citing Louis Kunkel, as his laboratory was responsible for the initial cloning of DMD and discovering the protein dystrophin, but the seminal articles are: (1) Monaco AP et al, Nature 1986;323:646-650 (initial cloning of DMD); (2) Koenig M et al, Cell 1987;50:509-517 (cloning of entire DMD gene); and Hoffman EP et al, Cell 1987;51:919-928 (initial discovery of the protein dystrophin). Some or all of these articles should be cited instead.

Response 1: Thank you for your comment. We have added all corresponding references.

Comment 2: Regarding the life expectancy into the 30s and 40s for DMD, that is correct but only in the context of good supportive care. Without such care, the life expectancy is much shorter and thus this caveat should be noted.

Response 2: Thank you for your important comment. We have changed the corresponding sentence as follows:

Lines 70-73: However, multidisciplinary care planning, including corticosteroids, cardiac medications, orthopedic surgery, rehabilitation, and assisted ventilation, has been found to improve quality of life and clinical outcomes, delaying death in patients with DMD into their 30s or 40s [10, 11]. 

Comment 3: The Introduction notes 2 antisense oligonucleotide drugs but the Discussion mentions 4. There are 4 such compounds approved in the US to date so the Introduction should be corrected.

Response 3: Thank you for your comment. The corresponding sentence has been changed as follows:

Lines 74-78: Recently, various therapies based on molecular mechanisms, including gene therapy, cell therapy, read-through drugs, and antisense oligonucleotides for exon-skipping, have been studied and subjected to clinical trials: of these, four antisense oligonucleotide drugs (eteplirsen, golodirsen, viltolarsen, and casimersen) have been approved by the United States Food and Drug Administration for use in treating dystrophinopathy [12-15].

Comment 4: I am a little confused about the composition of the UD cohort. The Methods section suggests that they are presymptomatic patients, and in the Results it suggests that the UD cohort has incomplete medical records. One or both of these passages should be clarified for consistency.

Response 4: Thank you for your proper comment. The corresponding phrase in the Methods has been changed as follows:

Lines 99-105: Five subgroups were defined, including DMD, BMD, intermediate phenotype muscular dystrophy (IMD), undetermined phenotype (UD) (patients who were ambulatory on medical records, but had no records after the age of 12 years) and symptomatic carriers (females with muscle weakness of any severity). DMD, BMD, and intermediate phenotype muscular dystrophy (IMD) were defined according to age at loss of ambulation (DMD <13 years, BMD ≥16 years, and 13 years ≤ IMD <16 years) in accordance with established diagnostic criteria [19].

Comment 5: It is best to avoid the term "wheelchair-bound", it is better to say "non-ambulatory" instead.

Response 5: Thank you for your comment. The corresponding wording has been changed as follows:

Lines 96-99: We analyzed the clinical spectrum of patients with dystrophinopathy using their medical records, which included information on age at symptom onset, sex, family history, motor weakness, presence of scoliosis, loss of ambulation, respiratory discomfort, dilated cardiomyopathy, and use of mechanical ventilation. 

Lines 99-105: Five subgroups were defined, including DMD, BMD, intermediate phenotype muscular dystrophy (IMD), undetermined phenotype (UD) (patients who were ambulatory on medical records, but had no records after the age of 12 years) and symptomatic carriers (females with muscle weakness of any severity). DMD, BMD, and intermediate phenotype muscular dystrophy (IMD) were defined according to age at loss of ambulation (DMD <13 years, BMD ≥16 years, and 13 years ≤ IMD <16 years) in accordance with established diagnostic criteria [19].

Lines 178-180: Loss of ambulation was significantly more frequent in patients with DMD (118 patients, 98%) and IMD (21 patients, 100%) than in patients with BMD (14 patients, 21%, p<0.001).

Lines 237-238: Most patients with DMD were non-ambulatory at the age of 10, used a mechanical ventilator at the age of 20, and were diagnosed with dilated cardiomyopathy at the age of 21.

Comment 6: In the methods, it is best not to say "We also assessed the serum creatine kinase..." and "Echocardiography was performed..." as the wording suggests that these data were collected prospectively. As this is a retrospective study, it is better to say that results of prior CK and echocardiography studies were extracted from the medical records.

Response 6: Thank you for proper comment. The corresponding word has been changed as follows:

Lines 110-112: The results of prior creatine kinase (CK), echocardiography studies, and muscle biopsies were extracted from the medical records in 227, 162, and 43 patients, respectively.

Comment 7: In the Genetic spectrum section, the first sentence should end with ", respectively."

Response 7: Thank you for proper comment. The corresponding sentence has been changed as follows:

Lines 185-186: Exonic deletions and duplications were recorded in 147 (67%) and 31 (14%) of the 218 unrelated probands, respectively.

Comment 8: The description of the 4 exonic duplications affecting two regions. I assume these are large duplications rather than two duplications for each patient but this should be clarified, perhaps even by indicating the start and end exons for the duplications.

Response 8: Thank you for comment. We have added the following sentences:

Lines 194-198: Exonic duplications affecting the two regions were found in four probands with DMD. Four patients with DMD (ID 139, ID 153, ID187, and ID 201) had duplications of exons 2 - 7 and 45 – 51, duplications of exons 5 - 37 and 50 – 59, duplications of exons 52 - 53 and 56 – 61, and duplications of exons 52-53 and 56-61, respectively (S1 table).

Comment 9: The genotype-phenotype correlation data is in line with the findings of other studies so I believe this information is largely correct. However, given that steroids prolong the ambulatory phase of the disease, the lack of steroid information potentially introduces errors as a patient who would ordinarily be classified as DMD may walk a bit past the age of 16 with steroid therapy. If possible, it would help to obtain steroid information. If this is not possible, at the very least this limitation should be expanded in the Discussion and also information should be provided on how widespread steroid use is in the DMD population in Korea.

Response 9: Thank you for very important comment. We agree with your opinion. 

The corresponding sentence has been changed as follows:

Lines 262-267: The use of steroids prolongs the ambulatory phase of dystrophinopathy. Therefore, the lack of steroid information may result in errors in distinguishing DMD from IMD and BMD. However, we could not confirm the exact drug history including steroids because that most of patients had also been treated at other hospitals. We could only estimate that about two-thirds of patients had been treated with steroids according to the experience of other myology center in Korea [18].

Comment 10: In Figure 2, it is difficult to interpret the exact CK levels at the later age ranges. It might be more informative to show two separate graphs, one for ages 0-12 or so and the other for ages 12 and above. Thus the second graph would be capped at 10,000 and it will be more clear what the later numbers are.

Response 10: Thank you for proper comment. Following your advice, we have corrected Figure 2. 

Comment 11: Figure 3 would be more informative if bars or lines were shown indicating the actual rearranged DNA segments.

Response 11: Thank you for proper comment. We have corrected Figure 3. 

 

[Reviewer 2]

The Authors report on 227 Korean patients from 218 unrelated families with dystrophinopathy; they describe retrospectively the clinical course of the disease. In all the patients they performed manual muscle testing (MMT) scores and CK dosage, only some patients underwent Echocardiography and muscle biopsy.

In all the patients was identified a pathogenic mutation in the dystrophin gene.

Comment 1: Clinical findings are compatible with those of previous reports; the Authors include an additional phenotype named undetermined phenotype (UD) for the patients who were not wheelchair-bound and too young to be defined as DMD or BMD. Regarding this group, in the supplementary table 1 18 patients are reported as “clinical phenotype pending”. Is the UD phenotype?

Response 1: Thank you for proper comment. “Clinical phenotype pending” means the UD phenotype. We have corrected the corresponding words in the S1 table. 

Comment 2: I think the group UD is confounding, if the muscle biopsy has been performed I would suggest to define the phenotype as BMD or DMD based on the immunohistochemical analysis; if not performed I would classify these patients referring to the phenotype expected based on the genotype. All the 18 mutations are reported in the LOVD pages and only the phenotype associated to the duplication of exon 14-45 it could be difficult to predict (transcript analysis is necessary). A part form this group, it would be interesting to add some information on the muscle biopsy performed in 43 patients and to correlate it with the genotype.

Response 2: Thank you for very important comment. We agree with your opinion. However, classifying subtypes based on multiple criteria may obscure the meaning of the subtypes, so we grouped them based on the age at loss of ambulation. Among 18 patients with UD phenotype, the results of muscle biopsy were extracted from the medical records in only one patient (ID16). Immunohistochemistry for the rod domain, C-terminus, and N-terminus of dystrophin protein showed a total loss of expression. The following sentences were added: 

Lines 156-159: Among all patients, 18 were classified as having UD phenotype. The results of muscle biopsy were extracted from medical records in only one patient (ID16). Therein, immunohistochemistry for the rod domain, C-terminus, and N-terminus of dystrophin protein showed a total loss of expression.

Comment 3: Regarding the genetic data the frequencies of exonic deletions/duplications and small mutations are similar to the ones previously reported in other countries. Do the Authors have cases of DMD confirmed at the biopsy (absence of dystrophin) and negative to molecular analysis?

Response 3: Thank you for critical comment. Of course, we have cases of dystrophinopathy confirmed at biopsy and negative to molecular analysis. However, our study is a retrospective observational study. Therefore, patients were not subjected to uniform genetic analysis, but to various testing, including PCR, MLPA, targeted sequencing, and whole exome sequencing. Actually, we excluded pathologically confirmed patients who were negative to MLPA but were not tested by next generation sequencing. 

This paper has some limitations as described by the Authors but it is important because is the first report on dystrophinopathies in Korean patients and give data for developing therapeutic drugs and designing clinical trials.

I have few comments :

Comment 4: Lane 37 “dystrophinopathies is a genetic muscle disease” Is not correct, is a group of inherited phenotypes all due to dystrophin mutations.

Response 4: Thank you for your comment. The corresponding sentence was changed as follows.

Line 36: Dystrophinopathy is a group of inherited phenotypes arising from pathogenic variants in DMD.

Line 55: Dystrophinopathy is a group of inherited phenotypes arising from pathogenic variants in DMD.

Comment 5: ABSTRACT and Lane 58 instead of “sequence variants” I’d rather say “mutations”

Response 5: Thank you for your comment. However, the terms “mutation” often lead to confusion because of incorrect assumptions of pathogenic and benign effects. Therefore, ACMG guidelines recommend that mutation is replaced by variant. We referred to small sequence variants, including single nucleotide variants and indels.

Comment 6: Lane 62 1% of deep intronic mutations is missing

Response 6: Thank you for your comment. However, we used small sequence variants, including indels and single nucleotide variants (nonsense, splicing, frameshift, missense, and deep intronic variants)

Comment 7: Lane 65 is the most common type of muscular dystrophy I would add “IN CHILDREN”

Response 7: Thank you for your comment. We added “in children.”

Line 62: Dystrophinopathy is the most common type of muscular dystrophy in children.

Comment 8: Lane 105 “undetermined phenotype (UD) (for the patients who were not wheelchair-bound and too young to be defined as DMD or BMD) “ it is not specificed the age considered as cut off

Response 8: Thank you for proper comment. The corresponding phrase was changed as follows.

Lines 99-102: Five subgroups were defined, including DMD, BMD, intermediate phenotype muscular dystrophy (IMD), undetermined phenotype (UD) (patients who were ambulatory on medical records, but had no records after the age of 12 years) and symptomatic carriers (females with muscle weakness of any severity).

Comment 9: Lane 219 “Nine unrelated probands did not follow the reading frame rule” A part from the patients with the del 3-7 for which the explanation is known, in the other patients it would be worth to study the transcript if the RNA from muscle biopsy is available.

Response 9: Thank you for very important comment. However, our study is a retrospective observational study. We will perform transcript analysis in future study.

---

## [Decision Letter · Decision Letter 1]

8 Jul 2021

Clinical and genetic spectra in patients with dystrophinopathy in Korea: A single-center study

PONE-D-21-16348R1

Dear Dr. Park,

We’re pleased to inform you that your manuscript has been judged scientifically suitable for publication and will be formally accepted for publication once it meets all outstanding technical requirements.

Kind regards,

Michael Kyba

Academic Editor

PLOS ONE

Additional Editor Comments (optional):

Thank your for addressing the reviewers' comments. Paper is good to go.

Reviewers' comments:

Reviewer's Responses to Questions

**Comments to the Author**

1. If the authors have adequately addressed your comments raised in a previous round of review and you feel that this manuscript is now acceptable for publication, you may indicate that here to bypass the “Comments to the Author” section, enter your conflict of interest statement in the “Confidential to Editor” section, and submit your "Accept" recommendation.

Reviewer #1: All comments have been addressed

Reviewer #2: All comments have been addressed

2. Is the manuscript technically sound, and do the data support the conclusions?

Reviewer #1: Yes

Reviewer #2: Yes

3. Has the statistical analysis been performed appropriately and rigorously? 

Reviewer #1: Yes

Reviewer #2: Yes

4. Have the authors made all data underlying the findings in their manuscript fully available?

Reviewer #1: Yes

Reviewer #2: Yes

5. Is the manuscript presented in an intelligible fashion and written in standard English?

Reviewer #1: Yes

Reviewer #2: Yes

6. Review Comments to the Author

Reviewer #1: The authors have responded constructively to the prior comments. The manuscript has been improved significantly. No further concerns.

Reviewer #2: (No Response)

7. PLOS authors have the option to publish the peer review history of their article (what does this mean?). If published, this will include your full peer review and any attached files.

Reviewer #1: No

Reviewer #2: No

---

## [Editor Report · Acceptance letter]

15 Jul 2021

PONE-D-21-16348R1 

Clinical and genetic spectra in patients with dystrophinopathy in Korea: A single-center study 

Dear Dr. Park:

I'm pleased to inform you that your manuscript has been deemed suitable for publication in PLOS ONE. Congratulations! Your manuscript is now with our production department. 

Kind regards, 

on behalf of

Dr. Michael Kyba 

Academic Editor

PLOS ONE